# Cholinergic modulation supports dynamic switching of resting state networks through selective DMN suppression

Pavel Sanda[1]*, Jaroslav Hlinka[1,2], Monica van den Berg[3,4], Antonin Skoch[2,5], Maxim Bazhenov[6‡], Georgios A. Keliris[3,7‡], Giri P. Krishnan [6,8‡]*

1 Institute of Computer Science of the Czech Academy of Sciences, Prague, Czech Republic, 2 National Institute of Mental Health, Klecany, Czech Republic, 3 Bio-Imaging Lab, University of Antwerp, Antwerp, Belgium, 4 µNEURO Research Centre of Excellence, University of Antwerp, Antwerp, Belgium, 5 MR Unit, Department of Diagnostic and Interventional Radiology, Institute for Clinical and Experimental Medicine, Prague, Czech Republic, 6 Department of Medicine, University of California San Diego, La Jolla, California, United States of America, 7 Institute of Computer Science, Foundation for Research and Technology - Hellas, Heraklion, Crete, Greece, 8 Georgia Institute of Technology, Atlanta, Georgia, United States of America

‡These authors are joint senior authors on this work.
* sanda@cs.cas.cz (PS); giri@gatech.edu (GPK)

**Data Availability Statement:** All relevant data are within the paper and its Supporting information files. The final code was deposited in ModelDB database, model no. 2016670. The model will be

## Abstract

Brain activity during the resting state is widely used to examine brain organization, cognition and alterations in disease states. While it is known that neuromodulation and the state of alertness impact resting-state activity, neural mechanisms behind such modulation of resting-state activity are unknown. In this work, we used a computational model to demonstrate that change in excitability and recurrent connections, due to cholinergic modulation, impacts resting-state activity. The results of such modulation in the model match closely with experimental work on direct cholinergic modulation of Default Mode Network (DMN) in rodents. We further extended our study to the human connectome derived from diffusion-weighted MRI. In human resting-state simulations, an increase in cholinergic input resulted in a brain-wide reduction of functional connectivity. Furthermore, selective cholinergic modulation of DMN closely captured experimentally observed transitions between the baseline resting state and states with suppressed DMN fluctuations associated with attention to external tasks. Our study thus provides insight into potential neural mechanisms for the effects of cholinergic neuromodulation on resting-state activity and its dynamics.

## Author summary

Brain activity during the resting period, commonly referred to as resting-state activity, is known to display coherent spontaneous fluctuations, which form various functional networks. Yet, the origin of these low-frequency (less than 0.05 Hz) fluctuations is poorly understood. In this work, we ask why is, in specific brain regions, such slow activity higher during rest periods compared to active periods. We use computational modeling and experimental data to demonstrate that the reduction of cholinergic release results in an

also included in the simulation environment of cybershuttle.org.

**Funding:** This work was supported by the Czech Science Foundation (pro-ject No. 21-32608S to PS and JH); by the MH CZ – DRO (NUDZ, 00023752 to JH and AS); by the Fund of Scientific Research Flanders (G048917N to MvdB and GAK); by the MH CZ – DRO (IKEM, IN: 00023001 to AS); by the National Science Foundation Grant (2209874 to MB and GPK) and by the National Institutes of Health (grant numbers RF1NS132913, 1R01MH125557, 1R01NS104368, 1R01NS109553 to MB and GPK). The funders had no role in study design, data collection and analysis, decision to publish, or preparation of the manuscript.

**Competing interests:** The authors have declared that no competing interests exist.

increase in resting-state activity and its functional connectivity in a prominent resting state network—the DMN (Default-mode network). Our work supports the hypothesis that cellular intrinsic and synaptic changes mediated by (chemical) neuromodulatory mechanisms contribute to the increases of low-frequency fluctuations during the resting state.

## Introduction

Resting-state fluctuations have been established as one of the fundamental properties of spontaneous brain dynamics observed across different species in neuroimaging studies [1–3]. However the neural mechanism underlying the origin of the fluctuations remains poorly understood. High amplitude fluctuations are the main contributor to emergent patterns of functional connectivity patterns which are used to define the underlying architecture of functional networks [4, 5]. Analysis of these networks received a great deal of attention [6] and connectivity changes in disease states are considered to reflect underlying pathologies [7].

One of the most intriguing observations is the increased spontaneous activity in specific brain areas during rest periods. Early observations identified the Default mode network (DMN) as a sub-network that has higher activation and functional connectivity during rest periods compared to task periods [8, 9]. The DMN is now regarded as a fundamental functional network activated during internal processing modes and deactivated when attention shifts to external tasks [10].

What governs the transitions between DMN-dominated rest states and DMN-suppressed attentive states is widely speculated but not yet clear [11–14]. One hypothesis states that chemical neuromodulation plays a critical role in dynamic transitions between functional networks [15–17]. Specifically, cholinergic activity has been suggested to influence the balance between rest and task-related brain activity [18–20]. The basal forebrain (BF) is a major source of acetylcholine (ACh) in the neocortex with broad yet specific projections [21–23]. Activity in BF closely matches the activity of the default mode-like network (DMLN, animal homologue of DMN, [24]) in animal models, and changes in co-activation of BF and DMLN was reported in early-stage rodent models of Alzheimer's Disease [25]. In fact, BF itself was suggested to be an inherent part of DMN [26, 27]. In a recent work from our group, we measured changes in resting-state fMRI following exclusive activation of BF cholinergic neurons in transgenic ChAT-cre rats using chemogenetics. In particular, the injection of a synthetic drug (clozapine-N-oxide, CNO) designed to activate DREADD receptors expressed on BF cholinergic neurons resulted in decreased spectral amplitude and functional connectivity in DMLN, a hallmark of DMN suppression [28]. While these studies provide evidence for cholinergic modulation of resting-state activity, the underlying neural mechanism has not yet been identified.

In this study, we identify a possible mechanistic explanation for the cholinergic modulation of resting-state activity. We used a large-scale biophysical *in-silico* network model of rat and human connectome. The advantage of cellular-level biophysical modeling is that we could model the influence of ACh neuromodulation as changes to ionic and synaptic currents. Cholinergic modulation is not fully understood and vary by cell type and region [29]. Here, we identified that excitability and recurrent connection as potential critical components of cholinergic modulation that impacts resting-state activity. We used experimental results of direct cholinergic neuron manipulation to constrain our model of the DMLN network. We first demonstrate that the computational model can capture the same changes in DMLN as observed during cholinergic modulation in rodents. We then extend these findings using a

computational model of human connectivity derived from diffusion-weighted MRI (DW-MRI). Finally, we demonstrate that a selective increase in cholinergic activation only in DMN results in the suppression of DMN activity and functional connectivity without modifying sensory networks.

## Results

We used a biophysical model of infra-slow resting-state fluctuations based on our previous work [30]. This model includes a network consisting of conductance-based excitatory and inhibitory neurons with realistic synaptic AMPA, NMDA, and GABA synaptic connections. In addition, dynamic variables corresponding to intra and extracellular ion concentrations, including $K^+$, $Na^+$, $Cl^-$, and $Ca^{2+}$ ions, were included. In this model, the slow variation of ion concentration in time, specifically extracellular $K^+$ ion concentration, allows for slow change in the neuron's excitability and firing rate, leading to fluctuations in the 0.02 Hz range similar to slow resting-state activity (Fig 1D–1F). In this current work, we extend our previous model to incorporate cholinergic modulation through direct manipulation of intrinsic and synaptic currents (see the next section).

For the network simulations, the connectivity between different brain regions was identified through structural imaging methods: DW-MRI (humans) and axonal tracing (rodents). On the global scale, the brain regions were connected with diffuse long-range connections (Fig 1B). The strength of the connections was proportional to the weight between regions of the global structural connectome (Fig 1A). On a finer scale, each modeled brain region was represented by a locally connected group of 50 excitatory neurons and 10 inhibitory interneurons (Fig 1C). In the case of rodent brain simulations, the connectome representing the structural connectivity of the rat's DMLN was derived from the NeuroVIISAS database of axonal tracing studies [31].

### Cholinergic modulation of DMN resting state in rodents

To constrain and validate the effects of ACh modulations in our model, we used data from our previous chemogenetic experiments in transgenic ChAT-cre rats [28]. This transgenic rat model allowed the selective targeting of cholinergic neurons in the BF (the primary source of ACh release to the neocortex) using designer receptors exclusively activated by designer drugs (DREADDs) [32]. In these experiments, Blood Oxygenation Level Dependent (BOLD) activity was measured in the resting state before and after injection of CNO, a synthetic chemogenetic activator of DREADD receptors that were expressed exclusively in BF cholinergic neurons, resulting in widespread ACh release in the projections areas of these cells. To control for any off-target effects, CNO was also injected in animals which did not express DREADDs (sham animals). A second control, which we use as a baseline, was performed by injecting vehicle (saline) in the DREADD expressing animals with both control conditions resulting in no significant effects in the BOLD and functional connectivity. On the other hand, injections of CNO in the DREADD animals (ACh release) resulted in decreases in BOLD amplitude and spectral profile as well as reductions of functional connectivity and the fractional amplitude of low-frequency fluctuations (fALFF, [33]) in DMLN (Fig 2A).

The computational simulations modeling the experiments were setup in the following way. The connectivity of rat DMLN was used and synaptic activity from each anatomical area was used to derive BOLD signals, see Fig 2B. We compared the simulation during baseline resting state and during high ACh levels that correspond to the experimental post-saline baseline and CNO conditions, respectively (Fig 2B). We monitored several dynamic variables of the model —including firing rate, the activity of $Na^+/K^+$ pump, extracellular $K^+$ levels, neuronal

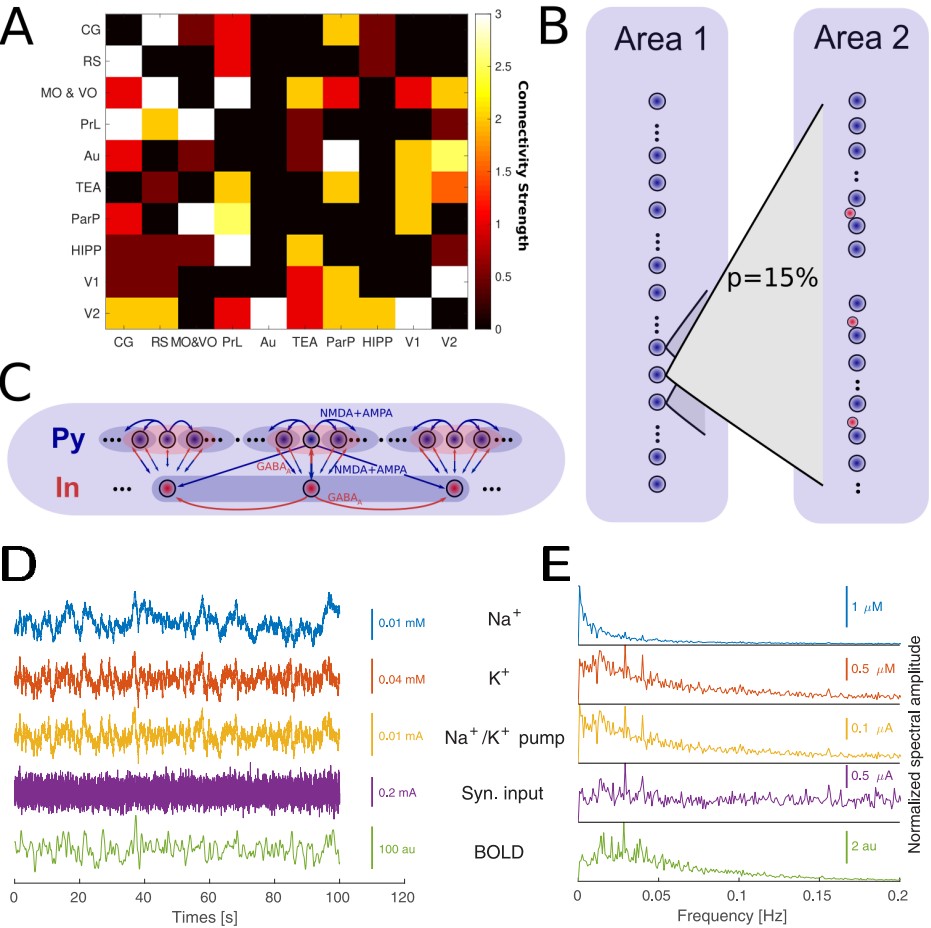

**Fig 1. Computational model.** A. Structural connectome of a rat's DMLN (imported from neuroVIISAS project). B. Model connectivity between two distinct DMN areas. Each source neuron has a p = 15% probability of connection to each neuron in the target area. The connectivity strength of AMPA connection between excitatory neurons was derived from the structural connectome in A. C: Model connectivity of excitatory and inhibitory neurons within a single module (area). D: Example traces of Na⁺ and K⁺ extracellular levels, the activity of Na⁺/K⁺ pump, average dendritic excitatory synaptic input (average across 50 excitatory neurons in a single area and 100 ms time window) and resulting BOLD trace (in arbitrary units). E. Spectral amplitude of the same variables as in E (single trial, average across all areas).

membrane voltage (spiking) and synaptic input. The total synaptic input was transformed into a BOLD signal representing the dynamics of each brain area (see Methods).

Past experimental work has shown that acetylcholine modulates several ionic currents and synaptic currents through nicotinic and muscarinic receptors. Based on these findings, we implemented a detailed model of cholinergic modulation as a reduction of somatic and dendritic potassium leak currents $I_K^{leak}$, somatic delayed-rectifier potassium current $I_{Kv}$, slowly activating potassium M-channel current $I_{Km}$, high-threshold Ca²⁺ current $I_{HVA}$, Ca²⁺–sensitive K + current $I_{KCa}$ in excitatory neurons based on past experimental work [34–37]. In inhibitory neurons, the somatic and dendritic $I_K^{leak}$ current and somatic $I_{Kv}$ current were reduced. Further, the influence of ACh on synaptic transmission was implemented as a decrease in excitatory AMPA connections and increase in NMDA connectivity based on experimental work [38–40]. By examining the impact of change in each of the ionic and synaptic currents on resting-state

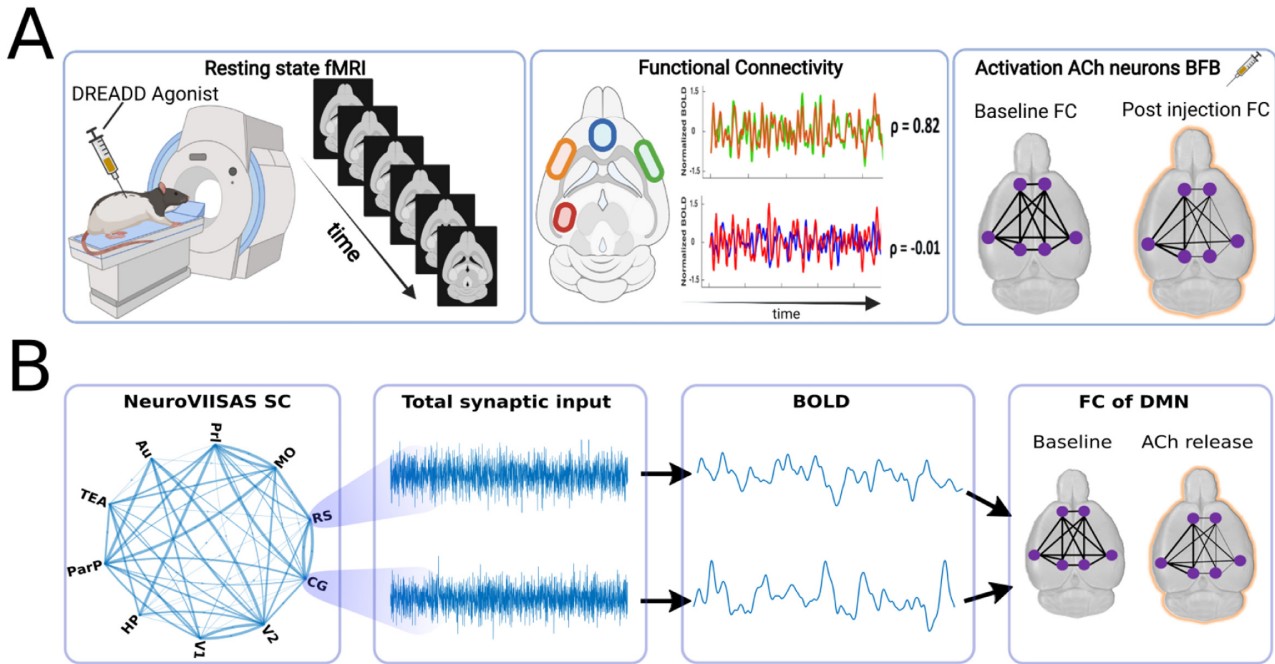

**Fig 2. Experimental and modeling methods.** A. Experimental framework. Chemogenetic tools (DREADDs) were used to selectively increase cholinergic activity in rat's basal forebrain (BF). Resting-state fMRI scans were performed during the resting state/after injection of saline and after the injection of CNO, resulting in upregulated cholinergic release in BF. Functional connectivity and other signal features were collected to compare both conditions. B. Simulation framework. The structural connectome of DMLN is the backbone of simulation dynamics. The total synaptic input of all neurons within each area is measured and used for conversion to a BOLD signal. The correlation of BOLD between areas was used to define functional connectivity of DMLN in two conditions—resting saline baseline and increased cholinergic release.

activity, we identified that K$^+$ leak and AMPA current had the largest impact on resting-state activity, see Fig 3 (detailed treatment of K$^+$ currents is in Fig A in S1 Text).

Adding cholinergic modulation to inhibitory neurons did not have large impact in our model (effects of modulating inhibitory population is shown in Fig B in S1 Text). This may be partly due to use of a canonical type of inhibitory neuron and future studies are required to

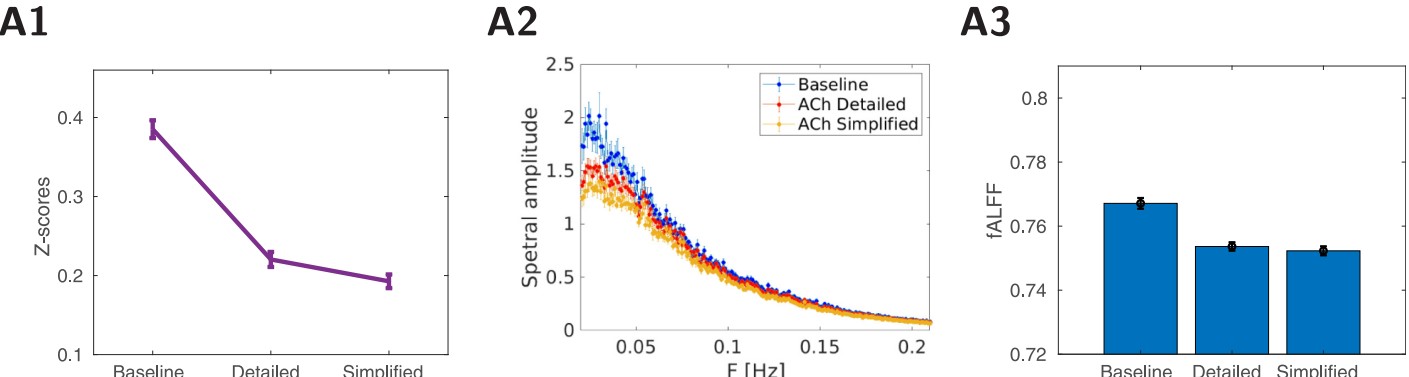

**Fig 3. Comparison of detailed model of cholinergic modulation and its simplified version.** A1: Average functional connectivity comparing baseline and ACh condition for detailed and simplified model (16 trials in each condition), Fisher z-transformed correlation ±SEM. B2: Average power-spectra from all DMLN areas (±SEM, 16 trials). The blue curve is the baseline condition, red/orange ACh modulation in detailed/simplified model. B3: fALFF values calculated from all DMLN areas. The bar graphs present mean fALFF values ±SEM.

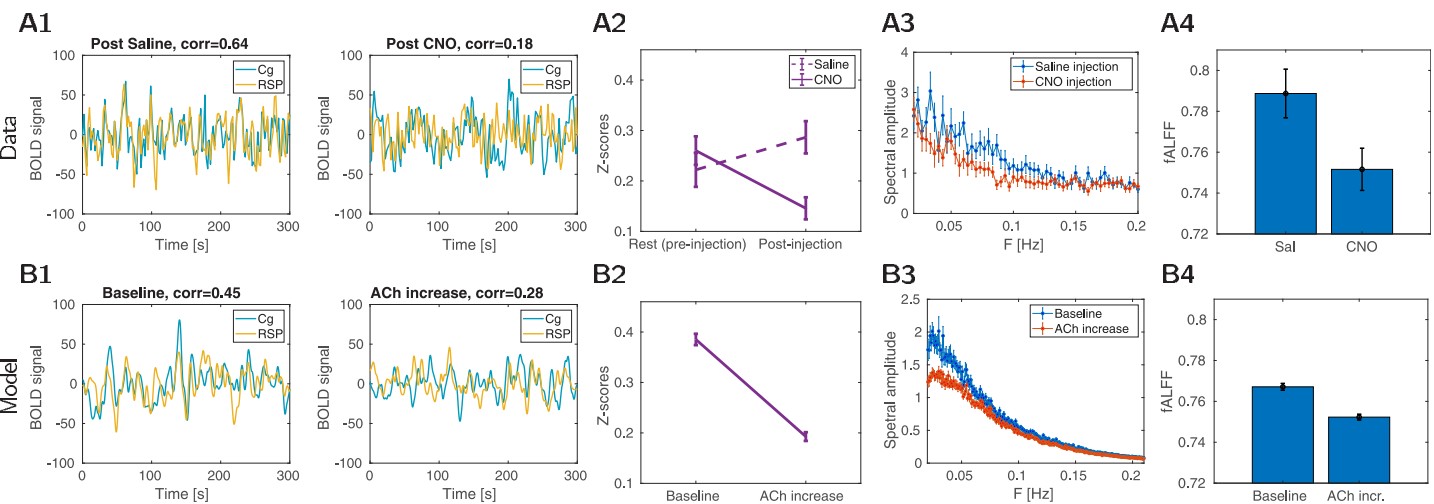

**Fig 4. BOLD properties.** Top row: Properties of BOLD signal in rats. A1: Example traces of BOLD in the cingulate and retrosplenial cortex. Left: post-saline condition ("baseline"), right post-CNO condition ("ACh activation"). A2: Average functional connectivity comparing resting control and condition 15–20 min after DREADD saline/CNO injection (16 animals in each condition), Fisher z-transformed correlation ±SEM. A3: Group-averaged power spectra of the seed-based maps of the cingulate in the right hemisphere (similar results can be obtained for the retrosplenial cortex). Blue curves are the power spectra after injection of saline; orange curves are the power spectra after injection of CNO. A4: fALFF values calculated from the seed-based FC maps of the right cingulate cortex (similar results can be obtained for retrosplenial cortex, see [28]). fALFF values were extracted from voxels of the right hemisphere after saline injection and CNO injection. The bar graphs present mean fALFF values ±SEM. Bottom row: Properties of BOLD in simulations. B1: Example traces of BOLD signal in the cingulate and retrosplenial cortex. Left: Spontaneous activity of the model ("baseline"), right: activity in the condition of ACh release. B2: Average functional connectivity comparing baseline and ACh condition (16 trials in each condition), Fisher z-transformed correlation ±SEM. B3: Average power-spectra from all DMLN areas (±SEM, 16 trials). The blue curve is the baseline condition, orange ACh condition. B4: fALFF values calculated from all DMLN areas. The bar graphs present mean fALFF values ±SEM.

implement different subtypes of inhibitory neurons (e.g. [41, 42]). All these findings allowed us to use a simplified model with changes to only K⁺ leak and AMPA currents in excitatory neurons and made possible the analysis of cellular mechanisms (detailed in the last section of the results). We use this simplified model in all subsequent sections.

After performing systematic variations of $K^+$ leak and AMPA conductance we identified that a reduction of 8% in $K^+$ leak currents and a 20% reduction in AMPA currents (local and long-range connections) qualitatively matched the results observed in DREADD experiments (i.e. changes in functional connectivity and spectral amplitude). The amplitude of the BOLD signal was reduced during elevated ACh condition in the model and experimental recordings (compare the top and bottom panels of Fig 4). The power spectrum in the low-frequency range (0–0.1 Hz) and fALFF measured across multiple trials were reduced with an increase in ACh. The experiment's spectral profile tended to have a wider frequency range (0–0.15 Hz) compared to the model (0–0.1 Hz). This difference is partly due to lower variability in the peak of resting-state activity in the model compared to the experiment. The average Z-score, which measures functional connectivity between all the regions, was also reduced following an increase in ACh. Similar reductions were also observed in $Na^+/K^+$ pump currents and fluctuations of extracellular $K^+$ concentrations. Overall, the change in excitability and recurrent connections due to ACh release in the DMLN model were sufficient to reproduce the changes in the BOLD signal of rat's DMLN following selective cholinergic neuron activation in BF.

We examined the correlation between the FC pairs of regions during saline condition in experiment (or rest in model) compared to CNO condition (ACh condition in model) (panels A,B in Fig F in S1 Text). In both cases, we observed a strong positive correlation between the two conditions. This suggests that there is a large influence of the baseline FC in both experiment and the model. The regions which are strongly coupled in saline (or rest) remained

strongly coupled following CNO (or ACh), but with lower strength. This correspondence demonstrate the same internal consistency in the qualitative nature of the change in FC following cholinergic modulation in model and experiment. However, due to low correlation between structural connection used in the model and FC during saline condition (panel C in Fig F in S1 Text), we did not observe significant correlation between model and experiment for the pairwise FC measure. This suggest that the SC we used in the model may lack additional details of the DMLN network—notably there is no full rat's connectome publicly available and thus interaction with the rest of the network is missing.

## Resting-state activity in a large-scale model of the human brain

In order to examine the cholinergic changes in humans, we first establish a baseline computational model with realistic structural connectivity based on human DW-MRI. The structural connections for the whole brain model were derived from the dataset of 90 healthy subjects (see Methods for details), describing the structural connectivity (SC) among brain regions defined by the AAL atlas [43]. While the extraction of structural connectivity matrix from DWI data generally faces a range of challenges [44] and may depend substantially on the particular method used, the current structural connectivity data have been thoroughly quality controlled and validated against an independent dataset (see [45] for details). The resulting average structural connectivity matrix for a single hemisphere is shown in Fig 5A1 (connectivity for both hemispheres and histogram of relative coupling strength between the brain areas is shown in Fig C in S1 Text). We observe typical properties of structural connectivity matrices extracted from DWI data. Namely, there is a skewed distribution of links with a small proportion of very strong links and a clustered structure with high density within functionally related brain areas. Further, the connectivity within the right and left hemispheres were very similar, giving rise to an almost symmetrical structure for the whole brain matrix. Because of this, together with the fact that inter-hemispheric connections are still imperfectly captured by current tractography methods, and higher computational demands, we used only a single hemisphere in our model and analysis (but see panel E in Fig C in S1 Text for an overview including both hemispheres). We also measured 15 mins of fMRI measurements in resting conditions from the same human subjects we used to estimate the structural connectivity; that allowed us to estimate functional connectivity. Fig 5A2 shows the average functional connectivity across subjects, with characteristic blocks of correlated regions. The relation between the mean SC and FC matrices is shown in Fig 5A3 (average correlation 0.50, for extended analysis see Table A in S1 Text) and is similar to other studies [46]. Of course, the observed SC-FC relation may differ depending on the specific pipelines used to estimate the SC and FC. For example, with more conservative preprocessing that aims to suppress potential artifact sources, the individual FC matrices resemble more the typical FC matrix [47] and, in our case, leads to globally decreasing strength of the functional connectivity (for the effects of commonly used preprocessing components see [48]); at the same time different fiber tracking methods would also lead to varying estimates of SC [49]. SC-FC correlations separately for weakly and strongly connected pairs is in Table A in S1 Text.

The BOLD activity from the computational model of the human connectome largely reproduced the functional blocks of FC (Fig 5A2 and 5B2) and the SC/FC relationship (Fig 5A3 and 5B3). We then examined how the model's FC compared with the experimental FC in strong and weak structural connections. Fig 5B1 shows the relationship between fMRI and the model's FC pairs—divided into two sets with strong (blue) and weak (red) structural connectivity. The separation in red versus blue points in this plot suggests that the model is able to capture the FC of strongly connected nodes better than weakly connected nodes (the visualization of

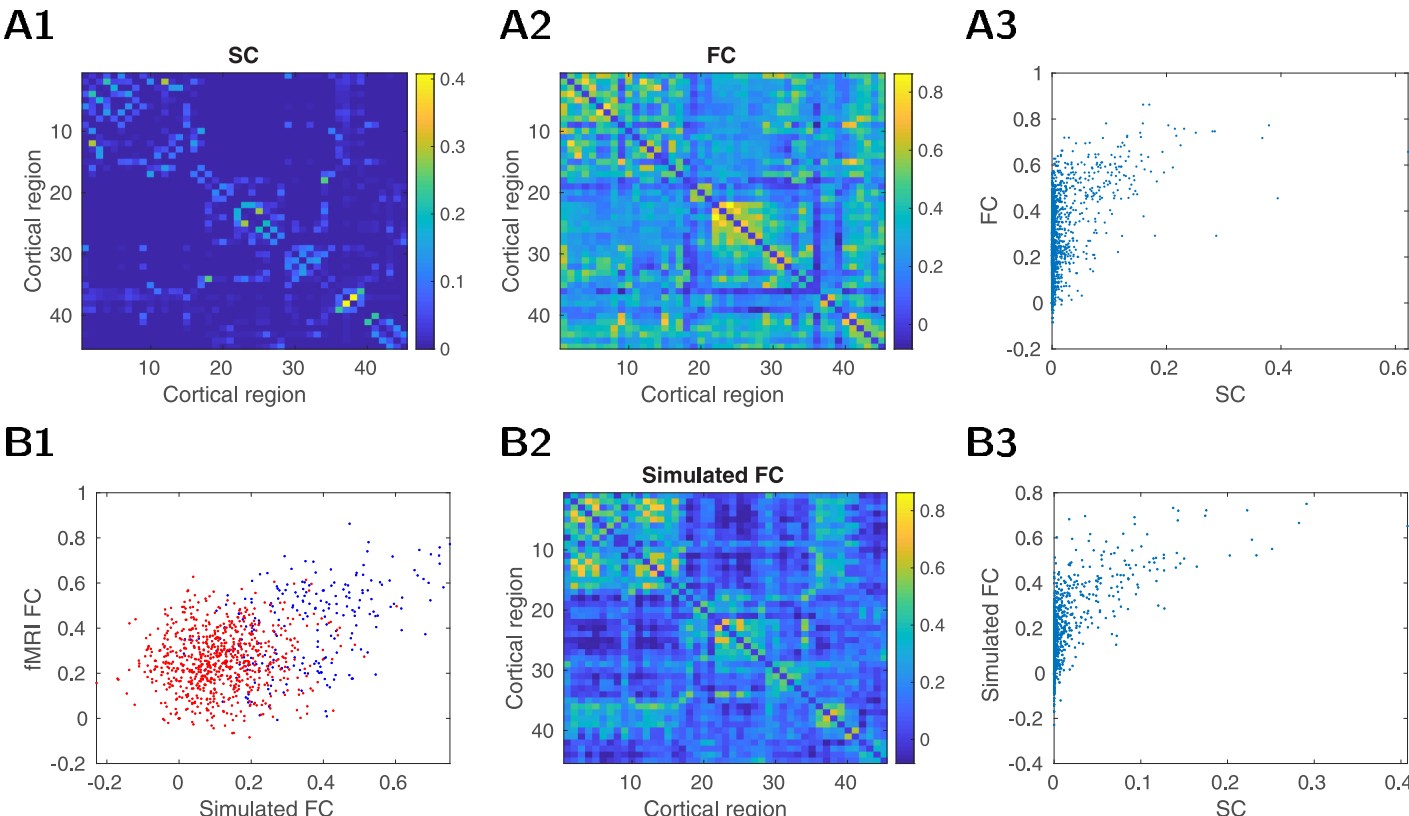

**Fig 5. Human model.** Top. Experimental human dataset. A1. Structural connectivity (SC), an average of 90 control subjects. The strength has an arbitrary scale. The mapping between numbers and their localization is in Table B in S1 Text. A2. Average functional connectivity (FC, 90 subjects). A3. SC-FC relationship avg. correlation was 0.50. Bottom. Model of resting state on the human connectome. B1. Relationship between experimental fMRI FC and FC of the modeled human connectome. The stronger connections (strength > 0.01, see SC above) have blue color. Avg. correlation 0.41 (stronger subset corr. 0.45, weaker subset corr 0.1). B2. BOLD FC of the model. The essential biophysical variables underlying the BOLD signal and their FC are visible in Fig E in S1 Text. B3. SC-FC relationship in the model, avg. correlation 0.63.

poorly performing ones on the FC matrix is shown in Fig D in S1 Text). The SC-FC correlation in our model is similar to other simulation studies [50]. Overall, our resting-state model using human connectome had the essential features often observed in human experiments and previous models.

## Cholinergic modulation influences global network properties

We next simulated the cholinergic modulation for the human connectome. Similar to rodent DMLN simulation cholinergic modulation was implemented by modifying $K^+$ leak currents and excitatory AMPA currents. Cholinergic modulation was first applied to all areas equally to simulate a broad ACh release across the brain (for more realistic case when ACh is not released uniformly across all regions see later sections). In this brain-wide high ACh condition, the amplitude of the resting-state activity measured by fALFF and the functional connectivity between regions decreased on average (Fig 6A and 6B). The reduction in fALFF, as well as in overall FC strength is consistent with rodent DMLN simulations in the previous section. In contrast, the SC-FC correlation increased with ACh (Fig 6C, panel D shows all SC-FC pairs). This increase was largely driven by a large reduction of FC in low SC ROI-pairs compared to

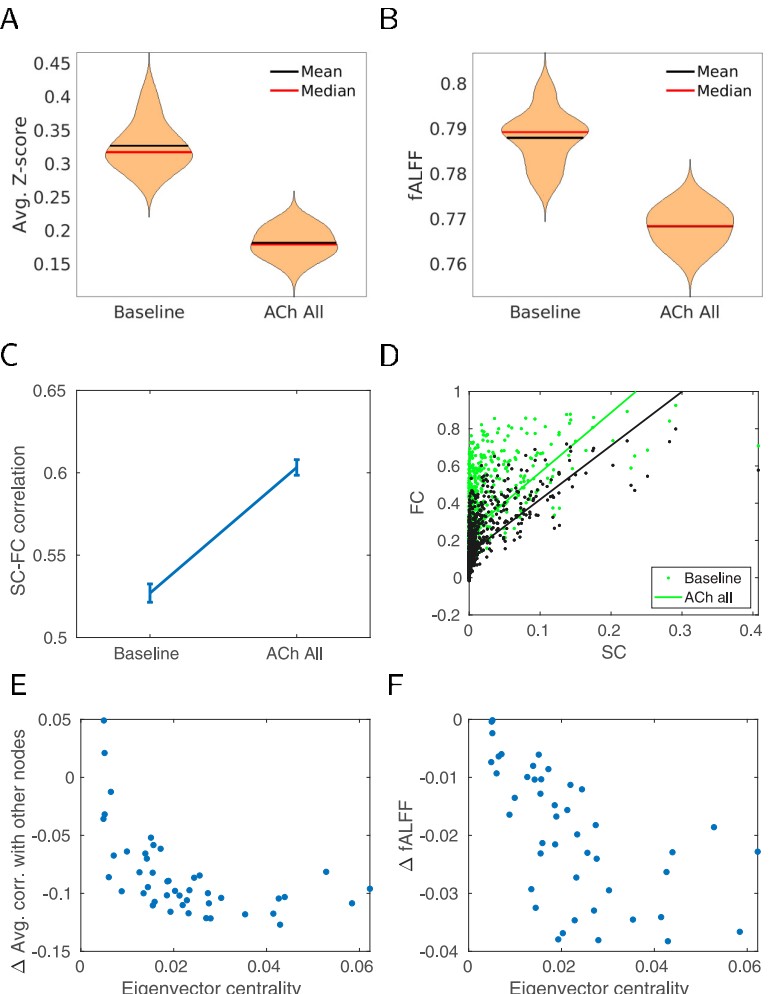

**Fig 6. Effect of generic cholinergic release on the connectivity (each condition 20 simulated trials).** A. Average functional connectivity of all pairs of nodes for baseline and cholinergic modulation (Z-scored, 20 trials). B. Average fALFF of all nodes for baseline and cholinergic modulation (20 trials). C. Average SC-FC correlation for baseline and cholinergic modulation, 20 trials ±SEM. D. SC-FC relation for each pair of nodes in the two conditions. E. Decrease of correlation with respect to node centrality. Each point represents one node, the x-axis corresponds to its eigenvector centrality of the node derived from the graph of structural connectivity. y-axis shows decrease of avg. correlation to all other nodes (for a given node). F. Decrease in amplitude (represented by fALFF) for each node of the network. x-axis as in E.

high SC ROI-pairs and is in line with expectations based on a linear approximation of the brain dynamics (see Discussion where we corroborate on this point).

A natural question arises concerning which brain regions should be most affected by the ACh modulation. For any given region there are two contributing factors that determine its response to cholinergic modulation. First, there is the reduction of resting-state activity fluctuations due to changes to excitability and recurrent connections. Second, the input to the region also changes from similar changes in other regions which are projecting to this region. In a network, these two factors interact leading to a cascading effect, with the strongest consequences for the highly connected regions receiving connections from other (highly connected) regions. This notion of overall network-propagated connectivity is conveniently captured in

the graph theoretical measure of eigenvector centrality [51] well studied in terms of resting-state activity [52].

We indeed observed that while the cholinergic modulation was applied uniformly, the nodes with higher eigenvector centrality had a larger reduction in resting-state amplitude (Fig 6F). As a consequence, these nodes also had lower FC for its connections. This suggests that the impact of cholinergic modulation depends on the connectivity, with central nodes being more impacted (and some of them residing in DMN). This selectivity further explains the impact of cholinergic modulation of SC-FC relationship.

## Differential impact of cholinergic modulation on DMN

We wanted to examine how cholinergic modulation impacts DMN sub-network which has been previously identified in resting-state studies. Previous work from our group [28] and others [19] suggests a larger influence of cholinergic modulation in DMN compared to task-positive network. One possibility is that DMN is more sensitive to cholinergic modulation, since regions of DMN are the major targets of cholinergic and non-cholinergeric projections of BF [53, 54]. In order to examine the selectivity of DMN, we examined the change in resting-state activity and FC under two different conditions: first, whole brain homogeneous ACh release, and second, ACh released only in the DMN areas of the brain ("DMN-only condition").

Fig 7A1 shows the average effect on FC in both conditions compared to the baseline. In both brain-wide and DMN-only conditions, there was a significant reduction of FC on average. The steepest decline is visible for the DMN nodes in the DMN-only condition. ACh increase in the entire brain resulted in a larger reduction of FC in DMN compared to DMN only condition and suggests that FC within DMN nodes is sensitive to changes in the rest of the brain.

We observed several intriguing findings when ACh was increased only in DMN. First, the largest change in FC was observed in the salience network (both within-FC and amplitude were reduced, Fig 7A2 and 7A3). This is expected as the saliency network shares 50% of nodes with DMN in AAL parcellation (see Table C in S1 Text and also compare SAL region of Fig 7B3 bottom vs. panel A2 bottom in Fig H in S1 Text). When DMN nodes were excluded from the salience network, changes in the rest of the saliency network became less prominent (see Fig G in S1 Text). More importantly, the sensory regions in auditory and visual networks were not impacted in the DMN-only condition. These findings demonstrate that selective cholinergic modulation of DMN has brain-wide changes in resting-state activity, with the notable exception of the sensory networks. If DMN activity is a marker of internal mentation and selective ACh release suppresses only DMN, it is favorable that the release does not suppress the sensory networks at the same time, so that antagonistic networks dealing with external-sensory environment can be decoupled/active. Thus, we hypothesize that selective cholinergic modulation could be one of the neural processes that facilitates the transition from internal to external oriented states (see Discussion for related experimental results).

We then examined the differences in the fine structure of FC across different conditions. Fig 7B show the p-value for each pair of nodes when comparing different conditions. We can see that in the case of whole brain ACh release (top triangle of the matrix), the change of FC generally follows block patterns of hubs seen in FC itself (compare Figs 7B1 top and 5B2), while DMN-only release modulates restricted part of the network (Fig 7B1 bottom triangle). Reordering the nodes in the matrix so that DMN nodes start first (thus DMN connectivity pairs form left top square matrix, Fig 7B2), we see that the DMN-only condition causes less global changes but was not limited to DMN regions. DMN itself can be roughly divided into two subsets of nodes—high and low responders to ACh change. We then used the eigenvector

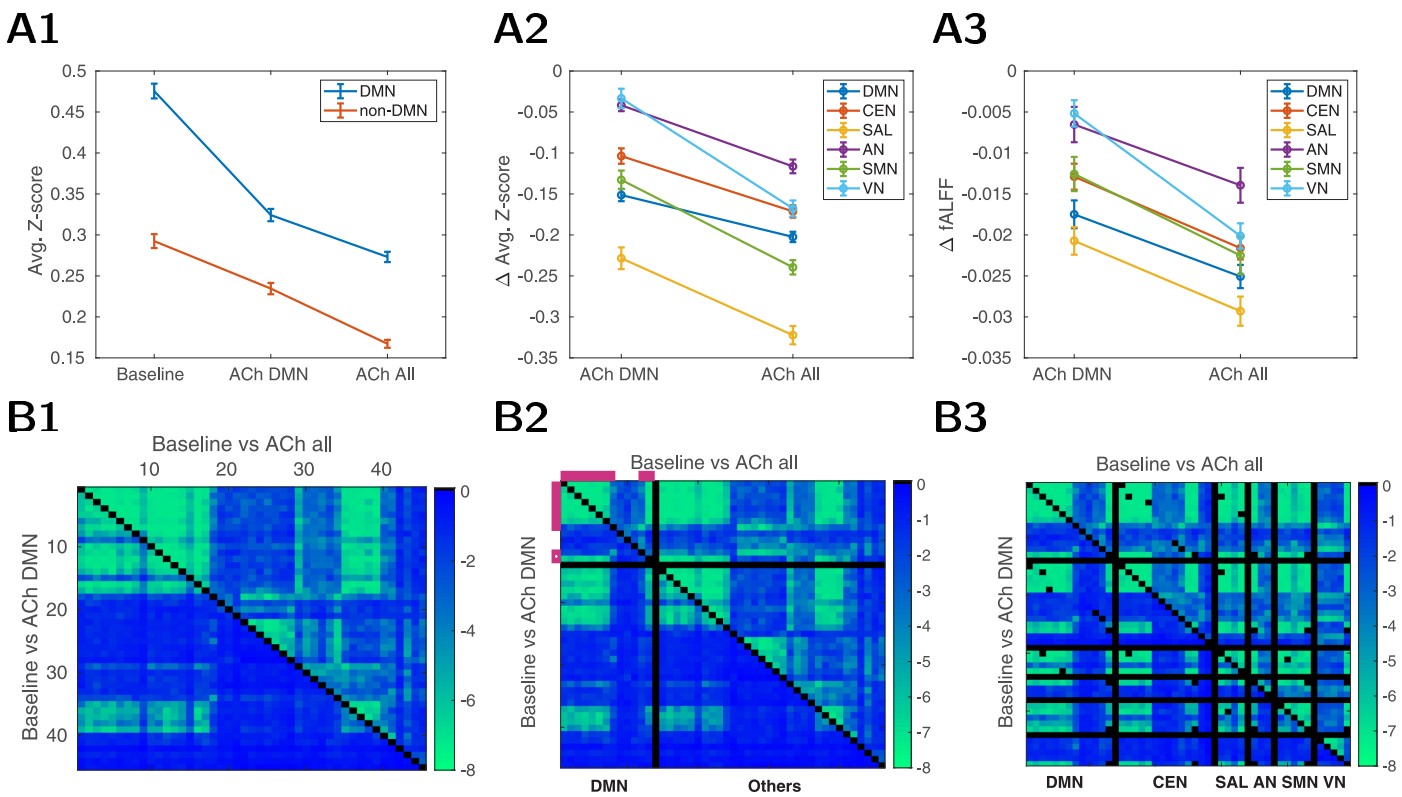

**Fig 7. Cholinergic modulation of the human connectome.** Three conditions (20 trials for each) are considered: baseline, ACh release only in DMN areas ("ACh DMN"), and ACh release in all brain areas ("ACh all"). A1: Average FC (Fisher z-transformed) computed separately within DMN (blue) and within the remainder of the areas (orange). A2: Changes in FC compared to baseline, projected to major functional networks (CEN = central executive, SAL = salience, AN = auditory, SMN = sensimotor, VN = visual network, for delineations, see Table C in S1 Text). A3: Changes in fALFFs compared to baseline, projected to major functional networks. B: FC changes after ACh release in all (top triangle) / DMN areas (bottom triangle). The color shows $\log_{10}$(p-value) of a two-sided Wilcoxon rank sum test, testing the null hypothesis that FC values in baseline and ACh condition are sampled from continuous distributions with equal medians (intuitively, the "blue regions" designate functional connectivities which are not substantially affected by the ACh modulation). Black color is used just as a separator. B1: Areas sorted as in AAL. B2: Areas regrouped to contain DMN and the rest of the areas separately. Violet color indicates DMN regions with higher (in upper 50% regions) eigenvector-centrality index. B3: Areas regrouped by their affiliation to different functional networks. Some nodes are shared across different networks (thus, some self-reference black dots out of the diagonal). For the B3 version without DMN nodes shared in other networks see panel A2/B2 in Fig H in S1 Text.

centrality as defined in the previous section and found that the nodes with high eigenvector-centrality index ("high influencers" indicated by violet color) had the largest change. The same can be stated as a general rule—high influencers show more ACh-related FC changes in the whole connectome when ACh targets all brain areas, see panel A1/B1, upper triangle in Fig H in S1 Text, where we sorted the nodes by their eigenvector-centrality rank.

To offer a comparable analysis to that shown for the rat experiments, we plot the effects of targeting only DMN by ACh in human connectome in a similar vein as in Fig 4. Again, we observed results matching the experimentally reported pattern [28], namely that DMN nodes under the ACh influence decreased the amplitude in the BOLD signal (Fig 8A and 8D), spectrum (Fig 8C) and functional connectivity (Fig 8B).

## Combined modulation of excitability and excitatory connections explain the state dependent change in resting-state activity

In order to better isolate the neural mechanism of cholinergic modulation of the resting state, we systematically varied the maximal conductance of $K^+$ leak current in excitatory neurons

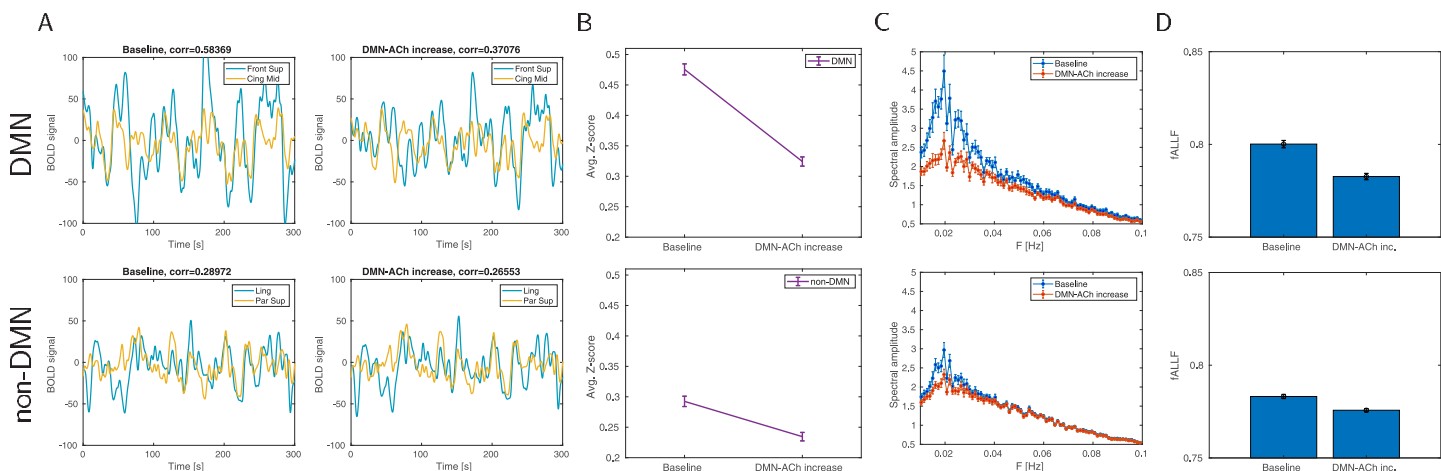

**Fig 8. The effect of ACh release in DMN on DMN and non-DMN areas of human connectome.** A. Example traces of BOLD signal. Top. Example of two areas in DMN —dorsolateral superior frontal gyrus and median cingulate/paracingulate gyri. Bottom: Example of two areas out of DMN—lingual gyrus and superior parietal gyrus. Left: Spontaneous resting activity of the model ("baseline"), right: the same trial when modulated by ACh release. B. Average functional connectivity comparing baseline and ACh condition (20 trials in each condition), Fisher z-transformed correlation ±SEM. Top: FC between DMN areas. Bottom: FC between non-DMN areas. C. Average spectral amplitude (±SEM, 20 trials) of all areas in DMN (top) and non-DMN areas (bottom) in resting condition (blue) and DMN-modulated-by-ACh condition (red). D. Average fALFF (±SEM, 20 trials) of all areas in DMN (top) and non-DMN areas (bottom).

and excitatory AMPA connections in a rat DMLN connectome (similar results were observed with human DMN). The baseline (or 100%) condition corresponds to the baseline (resting state) used in rat DMLN simulations. When the $K^+$ leak current conductance was reduced (from 110% to 85%), the mean firing rate across neurons increased from silence to 16 Hz (Fig 9A left and Fig 9B left). In contrast, AMPA conductance had a smaller impact on the firing rate, with the firing rate increasing moderately (3–4 Hz variation for lowest $K^+$ leak conductance). The higher sensitivity of $K^+$ leak conductance on firing rate is partly due to the impact of $K^+$ leak current on both direct excitability and the indirect effect through its influence on extracellular $K^+$ concentration. An increase in excitability due to the reduction of $K^+$ leak conductance also increases extracellular $K^+$ concentration (due to the accumulation of $K^+$ ions from spikes), which further increases the excitability. This feedback interaction, as reported in our previous models [55–57], may lead to a large non-linear change in firing rate with a change in $K^+$ leak conductance.

In contrast to the firing rate, the fluctuations in extracellular $K^+$ concentration and BOLD in resting-state frequencies were impacted by both $K^+$ leak and AMPA conductances. Its value increased with AMPA conductance only for the intermediate values (around 100%) of $K^+$ leak conductance. Specifically, the extracellular $K^+$ concentration doubled its value with AMPA conductance changed from 70% to 100% only for the intermediate values of $K^+$ leak conductance. The extracellular $K^+$ concentration fluctuation is lower at high $K^+$ leak conductance (above 102%) due to the significant drop in the excitability and the overall quiescence in the network activity. Interestingly, at low $K^+$ leak conductance values, when the firing rate is high, there is also the reduction in $K^+$ fluctuations due to smaller contribution of $K^+$ ions from $K^+$ leak currents and lower synchronization between regions (as shown by lower functional connectivity Fig 9A).

The BOLD, average functional connectivity (Fig 9A), and $Na^+/K^+$ pump activity (Fig 9C) closely matched the trend in the extracellular $K^+$ concentration. Both BOLD and $Na^+/K^+$ pump had the highest values for the intermediate values of $K^+$ leak and AMPA conductance. Further, the synchronization in BOLD (Fig 9A right) and $Na^+/K^+$ pump (Fig 9C) increased

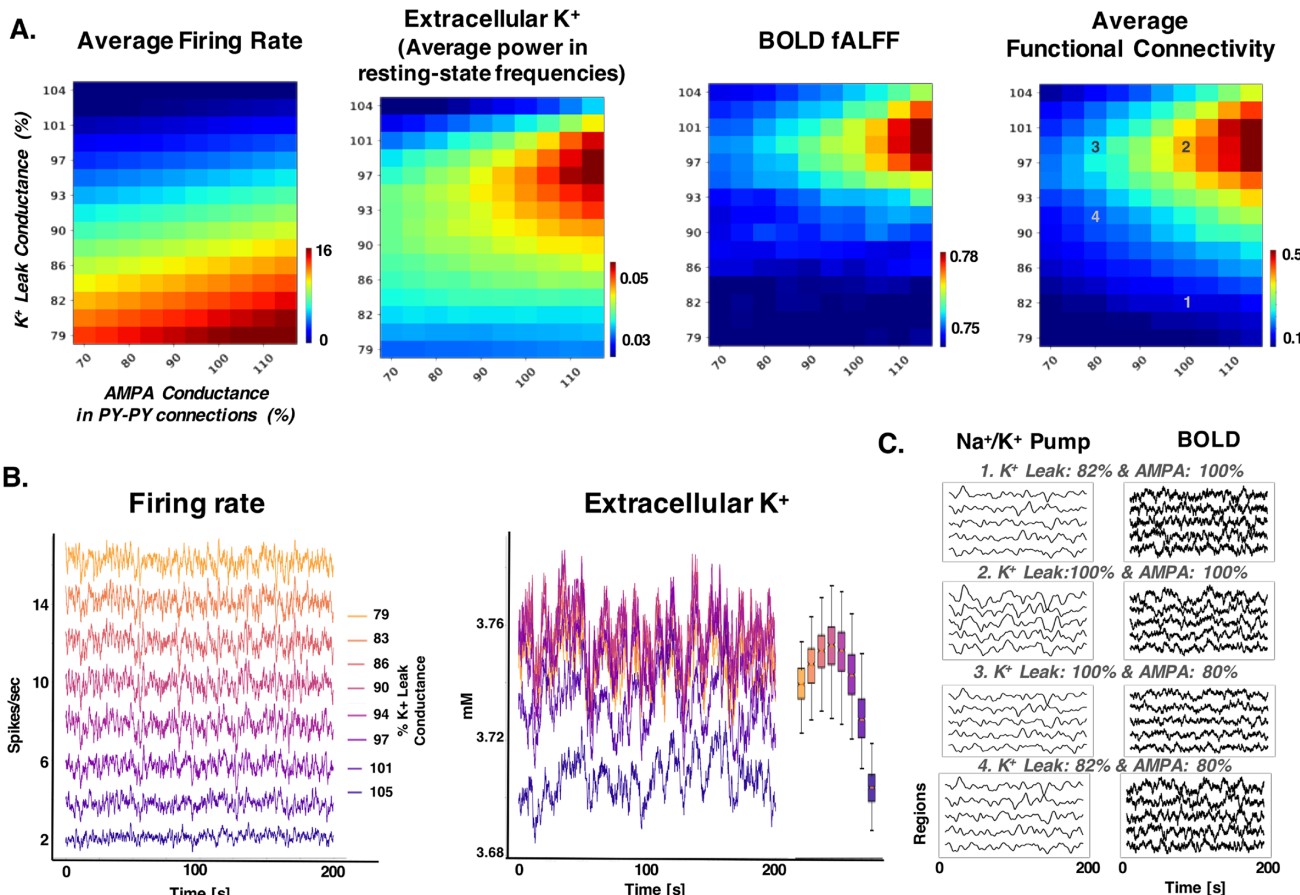

**Fig 9. Cellular mechanisms of cholinergic modulation of resting-state activity.** A. Different measures of resting-state activity and ion concentration dynamics when $K^+$ leak and AMPA conductances are varied. The average firing rate is measured by averaging the mean firing rate across neurons and regions. Extracellular $K^+$ concentration fluctuation is measured by taking the FFT of the average extracellular $K^+$ time series. BOLD fALFF, measured from the synaptic activity with fALFF similar to the method described in previous figures. Average functional connectivity is measured as the mean of functional connectivity across all pairs of regions. B. Time series of firing rate and extracellular $K^+$ for different values of $K^+$ leak conductance in 100% AMPA condition. The legend shows the correspondence between the color and value of the $K^+$ leak conductance value for both plots. Inset on the right of the extracellular $K^+$ plot is the boxplot with whiskers corresponding to the 1 and 99 percentile of the data to indicate the range of the extracellular $K^+$ fluctuations. C. $Na^+/K^+$ pump and BOLD time series for different regions for selected conditions. The number in the title corresponds to the number in the 2D sweep image in the right panel in A.

with AMPA conductance only in the intermediate range of $K^+$ leak conductance. The reduction in recurrent excitation led to lower BOLD, functional connectivity mediated by the changes to extracellular $K^+$ fluctuations (Fig 9C, panel 3). Thus, these findings suggest that the interaction between the excitability of neurons and the strength of recurrent excitation critically determines the fluctuations of $K^+$ in resting-state activity, $Na^+/K^+$ pump, and the BOLD. Further, our findings suggest that cholinergic modulation could also result in any of the intermediate values of $K^+$ leak and AMPA conductance, allowing for gradual and selective changes in resting-state activity.

Thus, cholinergic modulation which influences both excitability and recurrent excitation is ideally suited for modulating resting-state activity and its functional connectivity.

## Discussion

The widely used "resting-state" activity is thought to reflect intrinsic spontaneous activity when the brain is in rest periods [58]. While the general spatiotemporal patterns of rest and

task-related activity are surprisingly similar [59], some sub-networks, in particular the default mode network, have increased activity during rest periods as compared to a range of cognitive tasks [9, 60]. Moreover, during sustained demanding cognitive tasks, DMN activity fluctuation, while still apparent, is substantially reduced compared to the DMN activity fluctuation during an unconstrained resting state [61, 62]. In this study, we first tested the hypothesis that cholinergic activation promotes the inhibition of resting-state activity in DMN. To develop our model we used changes in rsfMRI following direct cholinergic neuron stimulation in rodents. We then extend the model to human connectome derived from DW-MRI measurements obtained from healthy human subjects. Results from our study support the hypothesis that a change in neuromodulation supports the switching between different functional networks.

Neuromodulation plays a critical role in the switch between different states of vigilance [63]. In addition, neuromodulation is also proposed to play an essential role in shaping the dynamics of the task- and resting-state networks [16, 17, 64]. Acetylcholine is a major neuromodulator released through broad projections from the basal forebrain [65]. We used data from a previous experiment [28], which involved selective activation of the cholinergic neurons in the basal forebrain using excitatory DREADDs. Simultaneously measuring resting-state fMRI with DREADDs showed suppression of rsfMRI during cholinergic neuron activation.

The results from the experiments were then used to constrain the computational model of DMLN with the connectivity based on NeuroVIISAS atlas for rodent brain [31]. We used a computational model of resting-state activity that included realistic ionic and synaptic currents based on the Hodgkin-Huxley formulation and ion dynamics. In this model, the interaction between cellular currents and ion dynamics leads to slow resting-state activity [30]. Using a biophysical model allowed us to implement cholinergic activation as a reduction of conductance of $K^+$ leak current and excitatory connections. The cellular action of acetylcholine is not fully understood and could be variable across neurons [29]. Thus, in this work we examined the most prominent action of cholinergic modulation involving $K^+$ leak and excitatory connections. We also examined a more elaborate model of cholinergic modulation involving all $K^+$, AMPA and NMDA currents, which resulted qualitatively similar to model involving only changes to $K^+$ leak and AMPA currents. The model was able to replicate the reduction in resting-state activity and its functional coupling following cholinergic activation.

An increase in DMN activity is a hallmark of internally oriented states and the transition between those states and states with externally oriented attention is poorly understood [10]. Several—not necessarily exclusive—candidates for the control of the transition are the activity of other functional networks [11, 66, 67], thalamocortical circuits, and ACh-dependent pathway mediated by basal forebrain (BF) [10, 24], which can be (together with the thalamus) thought of as a subcortical part of DMN [26]. Our results suggests that selective cholinergic modulation of DMN could facilitate this switch.

While it is known that BF projects broadly over the neocortex and the projections can be very specific in mammals [22, 23, 54, 68, 69], it is notoriously difficult to get exact human connectivity due to the small volume of the critical areas [70, 71]. We opt for the hypothesis that important DMN regions are more affected by cholinergic release (due to either higher probability in direct projections [72], or by the graded density of AChR receptors, or by a partial activity of specific regions of nucleus basalis of Meynert translating into partial neocortical activations). This is, of course, not the only possibility and there are more complicated accounts of how ACh acts globally [73] and within BF circuits [74–77]. As human experiments are limited to broad nicotine-related manipulations we initially show an animal model where we could directly influence major cholinergic center (BF), which in turn affects core hubs of

DMN—or more precisely, their animal DMLN analogues which were shown to be present across many species [78–83].

In this study, we only examined the cholinergic modulation arising from the basal forebrain. However, major projections of BF includes glutamatergic and GABAergic projections, which were not examined here. It has been proposed that glutamatergic and GABAergic projections mediate the BF influence on cortex [84] and in particular DMN [24, 27]. Specifically, there is an increase in glutamatergic input and a reduction of GABAergic input to cortex when the cholinergic neuron is not active [85]. Such an increase in glutamatergic input would further increase resting-state activity during rest periods in our model, and the overall results will be qualitatively consistent with the results reported in the current study. A more detailed computational model of BF subtypes could, in the future, isolate the cell type specific mechanisms.

Our model of resting state with human connectome captured several features of rsfMRI in humans, including the SC-FC relationship [86, 87]. The relation between SC and FC is far from straightforward, and while they are correlated, there is no simple match. Instead, there is a significant correlation variability reported across the studies [46]. However, our aim here was not to optimize SC-FC correspondence. First, this was already attempted in multiple simulation studies [50]; moreover, optimizing solely for the best match can even be detrimental to the models' dynamical properties [14]. Our model, however, shows a quantitative correspondence similar to the reported results. The functional connections directly supported by existing structural/anatomical connections were captured well, while weak structural connections rendered the prediction of functional coupling weak. The SC-FC match between the model and data could be improved if specific measures supporting cytoarchitectonic, transcriptomic, and higher order interactions were added to the model [88]. However, despite these additions, there remains a fundamental problem: human brain tractography is inherently limited and does not capture gray matter tracts and fibers going through thick bundles of axons, e.g. corpus callosum [89, 90]. Hence SC typically misses interhemispheric connections known to affect FC [91, 92]. Another problem stems from the fact that experimental FC values depend on specific parameters for preprocessing pipeline of the BOLD signal. Thus the predictive power of our model might not be directly comparable to other studies using different experimental SC/FC datasets.

We also observed that the SC-FC match increased with an increase in ACh (which generally reduces coupling between the nodes). This is in line with the expectations based on a simple linear approximation of the brain dynamics. In a linear process with weak coupling, the correlation matrix of the time series basically copies the structure of the coupling matrix, as only the first order interactions give rise to correlations of sufficient strength to be above the noise level. However, for a system with stronger coupling, the correlations due to indirect links (such as due to common source(s), or multiple steps of a causal chain) become strong enough to cause correlations (functional connectivity) above the noise level [93]. Thus, for strongly coupled systems, the functional connectivity can deviate further from the structural connectivity.

Few cases of indirect evidence suggest that SC-FC could be influenced by cholinergic modulation. It is known that FC change during task compared to resting-state [94] and that task-related activity is often accompanied by change in cholinergic modulation [95]. Assuming structural connections remains the same, it is likely the SC-FC also changes from resting-state to task. Further, some of the variability of SC-FC observed across subjects and studies [46] may be due to the state of subject and provide support to our observation. Finally, the reduction of SC-FC in Alzheimer's disease [96], where there is a reduction of cholinergic activation [97, 98] provide additional support to our findings. Relating SC-FC changes as a function of eigen-vector centrality is not common place in the literature but it might be worth to test our predictions on possibly related datasets (e.g. [99]).

In human connectome simulations, we concentrated on a particular role of ACh in switching the balance in the neocortical state from default mode network activation to activation of networks involved in external sensory processing [19, 100] or executive control [20]. DMN, a network active mainly in the resting condition [101], was associated with a variety of internally oriented mental states [102, 103] while inhibited with external goal-oriented tasks [8, 102, 104]. A recent experimental study [105] observed DMN suppression when participants transitioned from the rest to externally focused task and DMN activation during internal, self focused, task. At the same time there was a correlation between DMN and BF activity (source of ACh), suggesting a potential role of cholinergic modulation. In our simulations we considered global cholinergic modulation of the entire brain and selective cholinergic modulation only in DMN. Cholinergic modulation only in DMN regions translated to a picture consistent with the experimental results in rodents [28]; moreover, it showed DMN-specific inhibition which did not appear when ACh was uniformly affecting all cortical regions. Within the DMN, the modulation mainly affected the areas with higher eigenvector-centrality rank (i.e., highly connected nodes preferring connections to other highly connected nodes). As DMN richly connects (and even shares some nodes) with other functional networks, ACh-triggered changes in DMN propagate to other parts of the brain, however to a lesser degree than it is the case of uniform cholinergic release across all the areas; notably, the change did not impact internal coupling within visual/auditory networks. This observation makes cholinergic-related suppression of DMN compatible with independent activity in sensory regions connected with attention to external stimuli.

In conclusion our findings suggest cholinergic modulation on a cellular level leads to changes in large scale dynamics powerful enough to be a vital part of the intrinsic switching mechanism between different brain networks.

## Materials and methods

### Ethics statement

Human data. The study was conducted in accordance with the Declaration of Helsinki. The local Ethics Committee of the Prague Psychiatric Center approved the protocol on 29 June 2011 (protocol code 69/11). All participants provided written informed consent prior to their participation.

Animal data. All procedures were in accordance with the guidelines approved by the European Ethics Committee (decree 2010/63/EU) and were approved by the Committee on Animal Care and Use at the University of Antwerp, Belgium (approval number: 2015- 50).

### Biophysical model

The microcircuit connectivity and dynamics is identical to our previous work [30]. To briefly summarize, each network area (ROI) consists of 50 excitatory and 5 inhibitory neurons. Both excitatory and inhibitory neurons were modeled via axosomatic and dendritic conductance-based compartments following the equations

$$C_m \frac{dV_d}{dt} = -g_D^c(V_D - V_S) - I_D^{leak} - I_D^{pump} - I_D^{Int} \tag{1}$$

$$g_S^c(V_D - V_S) = -I_S^{leak} - I_S^{pump} - I_S^{Int}$$

$$I_D^{Int} = I_{Km} + I_{KCa} + I_h + I_{Ca} + I_{Na} + I_{NaP}$$

$$I_S^{Int} = I_{Na} + I_{Kv} + I_{Nap} + I_{KNa},$$

where $C_m$ is membrane capacitance, $V_{D,S}$ are dendritic/axosomatic compartment voltages, $g_{D,S}^c$ are leakage conductances, $I_{D,S}^{leak}$ are sums of the ionic leak currents, $I_{D,S}^{pump}$ are sums of Na$^+$ and K$^+$ currents through Na$^+$/K$^+$ pump, $I_{D,S}^{Int}$ are intrinsic currents. The dendritic compartment includes fast sodium current ($I_{Na}$), persistent sodium current ($I_{NaP}$), slowly activating potassium current ($I_{Km}$), calcium-activated potassium current ($I_{KCa}$), hyperpolarization-activated depolarizing mix cationic currents ($I_h$), high threshold Ca$^{2+}$ current ($I_{Ca}$) and leak currents [57, 106, 107]. The axosomatic compartment includes fast sodium current ($I_{Na}$), persistent sodium current ($I_{NaP}$), delayed-rectifier potassium current ($I_{Kv}$) and sodium-activated potassium current ($I_{KNa}$). Ion concentrations were modeled for intracellular K$^+$, Na$^+$, Cl$^-$, Ca$^{2+}$ and extracellular K$^+$, Na$^+$. K$^+$/Na$^+$ pump for K$^+$/Na$^+$ regulation and KCC2 cotransporter for extrusion of Cl$^-$ were used for both neuron types [57, 106, 108, 109].

Extracellular space was modeled for each neuron with local ion diffusion between nearest neighbors. It was tightly bounded between the glia and neurons, and there was an instantaneous and direct impact of ion concentration changes in the extracellular space on the neuronal and glial activity. Glial regulation of extracellular K$^+$ was modeled as a free buffer [57, 106, 107].

Local connectivity within single cluster was mediated via AMPA/NMDA conductances for PY->PY/IN and GABA$_A$ for IN->PY, local connectivity radiuses were $r(\text{Py} \rightarrow \text{Py}) \leq 5$, $r(\text{In} \rightarrow \text{In}) \leq 2$, $r(\text{In} \rightarrow \text{Py}) \leq 5$, $r(\text{Py} \rightarrow \text{In}) \leq 1$. Long range connections between clusters $i \rightarrow j$ was mediated via AMPA conductances with 15% probability of Py $\in i$ connecting to Py $\in j$, and GABA$_A$ conductances with restricted convergence of 5 Py $\in i$ to 1 In $\in j$ and probability of connection 25% for each possible connection.

Structural connectivity of human connectome was used (90 ROIs from AAL template, 45 for a single hemisphere, acquisition is described below), for modulation of DMN the subset of areas participating in default mode was defined by [110] (see Table C in S1 Text for explicit list of the nodes). Structural connectivity of rat DMLN (see Fig 1A) was extracted from the Neuro-VIISAS database of axonal tracing studies [31].

We use the synaptic currents as the primary source of BOLD [111, 112]. In particular, BOLD signal was created for each cluster first by averaging total synaptic input for all excitatory neurons in 100 ms windows and then it was convolved with a hemodynamic response function imported from Statistical Parametric Mapping (SPM) package [113]. In addition, we also observed correlation between the extracellular K+, Na+/K+ pump and the estimated BOLD activity (Fig E in S1 Text). The Na$^+$/K$^+$ pump activity reflect the metabolic activity of a given region (oxygen consumption) and this correlation further provide evidence that the BOLD activity measured in the model could capture the hemodynamics which is responsible for the BOLD activity in fMRI.

We performed various (1-D or 2-D) sweeps which revealed the qualitative nature and range of parameter values that best matched the results from experiments. The simple sweeps (see Fig A in S1 Text) showed that modulation of all K$^+$ currents modulation resulted in an inverted U-shape dependency for both functional connectivity and amplitude. A similar exploration was performed with inhibitory connections (see Fig B in S1 Text). Combining modulation of all K$^+$ current and AMPA conductance resulted in similar qualitative pattern, with the inverted U-curve slightly shifted (the same was later observed with simplified model in 2-D sweep of Fig 9). The parameters corresponding to firing rate around 5 Hz matched the peak of the inverted U-curve in the resting-state activity amplitude, which were taken as the 100% or rest condition. As the cholinergic modulation is known to reduce K$^+$/AMPA channel conductance, the parameters that were left of the peak and close to experimental FC/amplitude were taken as cholinergic condition. Admittedly this procedure does not examine the whole space

of parameter values, but the qualitative patterns observed were robust across variations of other parameters, consistent with cellular electrophysiological studies on cholinergic modulation and provide a hypothesis that can be tested in future experiments.

## Human data—Structural connectivity

Acquisition of MRI data and construction of structural connectivity was identical to the methods described in [45]. To summarize, the data provided here are based on MRI scans of 90 healthy control individuals participating in the Early-Stage Schizophrenia Outcome study [114]. The construction of structural connectivity matrices was based on a connectome generated by probabilistic tractography on diffusion MRI data. We used ROIs from the widely used AAL atlas (Automated Anatomical Labeling atlas, [43]). The connectivity between two ROIs is based on the number of streamlines in the tractogram beginning in one ROI and terminating in the other ROI. Accurate mapping of the AAL atlas ROIs to the diffusion data space was realized as a two-stage process: affine mapping of structural T1 images to MNI space and a rigid-body mapping between the T1 structural data and the DWI data, both for each subject.

We performed the MRI scanning at the Institute for Clinical and Experimental Medicine in Prague, on a 3 T Trio Siemens scanner (Erlangen, Germany). A 12-channel head coil was used, software version syngo MR B17. DWI data were acquired by a Spin-Echo EPI sequence with TR/TE = 8300/84 ms, matrix 112 × 128, voxel size 2 × 2 × 2 mm3, b-value 0 and 900 s/mm2 in 30 diffusion gradient directions, 2 averages, bandwidth 1502 Hz/pixel, GRAPPA acceleration factor 2 in phase-encoding direction, reference lines 24, prescan normalize off, elliptical filter off, raw filter on—intensity: weak, acquisition time 9:01. T1 3D structural image was acquired by using the magnetization prepared rapid acquisition gradient echo (MPRAGE) sequence with (TI—inversion time) TI/TR/TE = 900/2300/4.63 ms, flip angle 10˚, 1 average, matrix 256 × 256 × 224, voxel size 1 × 1 × 1 mm3, bandwidth 130 Hz/pixel, GRAPPA acceleration factor 2 in phase-encoding direction, reference lines 32, prescan normalize on, elliptical filter on, raw filter off, acquisition time 5:30.

## Human data—Functional connectivity

**fMRI data acquisition.**   Scanning was performed with a 3T MRI scanner (Siemens Magnetom Trio) located at the Institute for Institute of Clinical and Experimental Medicine in Prague, Czech Republic. Functional images were obtained using T2-weighted echo-planar imaging (EPI) with blood oxygenation level-dependent (BOLD) contrast using SENSE imaging. GE-EPIs (TR/TE = 2000/30 ms, flip angle = 70˚) comprised of 35 axial slices acquired continuously in sequential decreasing order covering the entire cerebrum (voxel size = 3×3×3 mm, slice dimensions 48x64 voxels). 400 functional volumes were used for the analysis. A three-dimensional high-resolution MPRAGE T1-weighted image (TR/TE = 2300/4.63 ms, flip angle 10˚, voxel size = 1×1×1 mm) covering the entire brain was acquired at the beginning of the scanning session and used for anatomical reference.

**Data preprocessing, brain parcellation, and FC analysis.**   The rsfMRI data were corrected for head movement (realignment and regression) and registered to MNI standard stereotactic space (Montreal Neurological Institute, MNI) with a voxel size of 2×2×2 mm by a 12 parameter affine transform maximizing normalized correlation with a customized EPI template image. This was followed by segmentation of the anatomical images in order to create subject-specific white-matter and CSF masks. Resulting anatomical images and masks were spatially normalized to a standard stereotaxic MNI space with a voxel size of 2×2×2 mm.

The denoising steps included regression of six head-motion parameters (acquired while performing the correction of head-motion) and the mean signal from the white-matter and

cerebrospinal fluid region. Time series from defined regions of interest were additionally filtered by a band-pass filter with cutoff frequencies 0.004--0.1 Hz. The regional mean time series were estimated by averaging voxel time series within each of the 90 brain regions (excluding the cerebellar regions) comprising the Automated Anatomical Labeling (AAL) template image [43]. To quantify the whole-brain pattern of functional connectivity, we performed a ROI-to-ROI connectivity analysis and computed for each subject the Pearson's correlation matrix among the regional mean time series, as (linear) Pearson's correlation coefficient has been shown to be suitable for fMRI ROI functional connectivity estimation [115].

### Animal data

Experimental framework and acquisition of MRI data is identical to [28]. To summarize, 28 adult ChAT-Cre Long Evans rats were used, of which 14 males and 14 females. Animals were group housed with a 12h light/dark cycle and with controlled temperature (20–24°C) and humidity (40%) conditions. Standard food and water were provided ad libitum.

All rats received stereotactic surgery targeting the right nucleus basalis of Meynert, horizontal diagonal band of broca and substantia innominata to transfect cholinergic neurons using either a Cre-dependent DREADD virus (AAV8-hSyn-DIO-hM3Dq(Gq)-mCherry) (N = 16) or sham virus (AAV8-hSyn-DIO-mCherry) (N = 12). Resting-state functional MRI was performed at least two months after surgery, to allow stable expression of the virus. Animals were anesthetized using a combination of isoflurane (0.4%) and medetomidine (0.05 mg/kg bolus followed by a 0.1mg/kg/hr continuous infusion). An intravenous catheter was placed in the tail vein which was used to administer 1 mg/kg CNO or saline. A gradient-echo EPI sequence was used (TE: 18 ms, TR: 2000ms, FOV: (30 x 30) mm2, matrix [128 x 96], 16 slices of 0.8 mm) on a 9.4T Bruker Biospec preclinical MRI scanner. A 5 minute baseline scan was followed by the injection of either CNO or saline during a 20 minute scan, followed by another 5 minute rsfMRI scan. DREADD expressing animals received two scan sessions, one with an injection of CNO and second session with an injection of saline, while Sham animals only received one scan session with CNO.

Preprocessing of the rsfMRI data included realignment, spatial normalization to a study specific template, masking, smoothing and filtering (0.01–0.2 Hz) using Matlab 2014a and SPM12 software [113]. Region-of interest based analysis was performed using predefined regions belonging to the default mode network. Amplitude of low frequency fluctuations were extracted from seed regions (cingulate cortex and retrosplenial cortex) within the right hemisphere. FC and fALFF values were compared before and after injection of CNO/saline using paired two-sample t-tests or unpaired two-sample t-tests.

## Supporting information

**S1 Text. Supplementary information file.** It includes supplementary figures A,B,C,D,E,F,G,H and supplementary tables A,B,C.
(PDF)

## Author Contributions

**Conceptualization:** Pavel Sanda, Jaroslav Hlinka, Georgios A. Keliris, Giri P. Krishnan.

**Data curation:** Pavel Sanda, Monica van den Berg, Antonin Skoch.

**Formal analysis:** Pavel Sanda, Jaroslav Hlinka, Monica van den Berg, Antonin Skoch, Georgios A. Keliris, Giri P. Krishnan.

**Funding acquisition:** Jaroslav Hlinka, Maxim Bazhenov, Georgios A. Keliris.

**Investigation:** Pavel Sanda, Monica van den Berg.

**Methodology:** Pavel Sanda, Jaroslav Hlinka, Monica van den Berg, Georgios A. Keliris, Giri P. Krishnan.

**Resources:** Maxim Bazhenov, Georgios A. Keliris.

**Software:** Pavel Sanda.

**Supervision:** Jaroslav Hlinka, Maxim Bazhenov, Georgios A. Keliris, Giri P. Krishnan.

**Validation:** Pavel Sanda, Monica van den Berg.

**Visualization:** Pavel Sanda, Giri P. Krishnan.

**Writing – original draft:** Pavel Sanda.

**Writing – review & editing:** Pavel Sanda, Jaroslav Hlinka, Monica van den Berg, Antonin Skoch, Maxim Bazhenov, Georgios A. Keliris, Giri P. Krishnan.

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
