## [Decision Letter · Decision Letter 0]

30 Oct 2023

Dear Dr Krishnan,

Thank you very much for submitting your manuscript "Cholinergic modulation supports dynamic switching of resting state networks through selective DMN suppression" for consideration at PLOS Computational Biology.

As with all papers reviewed by the journal, your manuscript was reviewed by members of the editorial board and by two independent reviewers. In light of the reviews (below this email), we would like to invite the resubmission of a significantly-revised version that takes into account the reviewers' comments.

We cannot make any decision about publication until we have seen the revised manuscript and your response to the reviewers' comments. Your revised manuscript is also likely to be sent to reviewers for further evaluation.

Sincerely,

Matthias Helge Hennig, Ph.D.

Academic Editor

PLOS Computational Biology

Marieke van Vugt

Section Editor

PLOS Computational Biology

Reviewer's Responses to Questions

**Comments to the Authors:**

Reviewer #1: The authors investigate the neural mechanisms underlying the modulation of resting-state brain activity by cholinergic neuromodulation. Using a computational model, the authors demonstrate that changes in excitability and recurrent connections influenced by cholinergic modulation impact resting-state activity. The results align with experimental findings in rodents regarding cholinergic modulation of the Default Mode Network (DMN). The study also extends its scope to human resting-state simulations, revealing that increased cholinergic input leads to a reduction in functional connectivity across the brain. Moreover, selective cholinergic modulation of the DMN mirrors observed transitions between baseline resting state and states with suppressed DMN fluctuations associated with attention to external tasks.

This model is useful tool to explore the functioning of neuromodulation mechanisms and sub-networks activations in the brain. For example, the authors conducted interesting experiments on neuromodulation of specific sub-networks and investigated the role of modulating specific parameters (e.g. K+ leak conductance and PY-PY AMPA conductance).

This research provides valuable insights into the neural mechanisms underlying the effects of cholinergic neuromodulation on resting-state brain activity and its dynamics, and on transitions between states with externally and internally oriented attention, which are still poorly understood.

Given the computational nature of this study, it's crucial for the authors to achieve a reliable validation of the results.

My major concern is that some observations on the model might not be easily confirmed through experiments, and it might not be easy to confirm their significance.

The tested hypothesis on neuromodulation could, in principle, work by chance, and not be the correct mechanism happening in the brain. Also, it could be distorted by the difficulty to constrain the model (e.g. given the experimental limitations in recording human connectome).

Major:

-line 77: “We used experimental results of direct cholinergic neuron manipulation to constrain our model of the DMLN network.” However, it is not clear to me exactly how the model is constrained. Maybe the authors could discuss this in more detail? To have a reliable tool to constrain models on experimental data is an open issue, and to demonstrate this could be a strength point of the work.

-I have a general concern how what is the qualitative evaluation of the model. E.g. to justify the validation of the model, the authors write: We then used a computational framework that sufficiently mimicked the experiments (line 127). But it is not clear to me if there is a quantitative evaluation of this comparison. They refer to Fig 9 in SI, but this is an important point that should be better discussed in the main text.

Regions decorrelate, after achetilcoline injection (simulation of increased excitability). But do they decorrelate in a similar way? Are there regions that decorrelate more than others? To be constructive, I would suggest measuring the correlation between the change in FC in model and simulations.

-In fig5 the change in FC is shown in the model, but there is no comparison with the data. This is due to the lack of human data for that condition. However, is it possible to validate the result in some way? For example, the increase of correlation between FC and SC, is it observed also experimentally in some condition?

-Same for the change as a function of eigenvector centrality. Can this be compared with any experimental finding?

-line 231: ‘Thus a selective cholinergic modulation of DMN would support the switch between externally oriented task states to internally oriented states’. For my understanding, the authors are saying that DMN only modulation does not affect sensory networks, but I don’t understand how this can demonstrate the link to the internally oriented states. Could the authors discuss better this point?

Minor:

-If I am not wrong, CNO is not introduced when firstly mentioned.

-Fig3, it could be made explicit in the figure that the two lines refer to data and model (maybe it could be explicitly written on the left of the panels, or using a thumbnail).

-In fig5 caption it might be more explicit that this is a results on simulation.

Reviewer #2: The authors have tried to delineate the role of cholinergic neuromodulation on brain activity during resting state by a combination of integrating experimental data and simulations in rodents and human subjects. The mechanisms of why low-frequency fluctuations in brain activity are greater particularly during resting periods as opposed to active periods are investigated.

A dominant resting state sub-network called the default-mode network (DMN) is described, which shows an increase in functional connectivity upon a reduction in cholinergic release, which also results in an increase in resting state brain activity. The authors further demonstrate using human resting-state simulations that brain-wide functional connectivity is reduced with increase in cholinergic input.

The study utilizes a biophysical network model of the rat and human connectome. In particular, the authors crucially demonstrate using computational modelling that changes in cellular excitability (cellular intrinsic properties) and recurrent connectivity (synaptic changes) by cholinergic modulation results in greater low-frequency fluctuations in brain activity during resting periods.

Some points that need more clarity:

1. The model has excluded the influence of cholinergic modulation on different cell types and focused primarily on excitatory connections. Is this oversimplified and leading to a biased result in the modeling?

2. The authors use resting state fMRI data as a measure of neuronal activity. It would be good to describe how fMRI data actually correlates to this, in comparison with cellular-level and large-scale neuronal population-level activity by other methods?

**Have the authors made all data and (if applicable) computational code underlying the findings in their manuscript fully available?**

Reviewer #1: **No: **data and code were not provided.

Reviewer #2: Yes

PLOS authors have the option to publish the peer review history of their article (what does this mean?). If published, this will include your full peer review and any attached files.

Reviewer #1: No

Reviewer #2: No
---

## [Decision Letter · Decision Letter 1]

23 Apr 2024

Dear Dr Krishnan,

We are pleased to inform you that your manuscript 'Cholinergic modulation supports dynamic switching of resting state networks through selective DMN suppression' has been provisionally accepted for publication in PLOS Computational Biology.

Best regards,

Matthias Helge Hennig, Ph.D.

Academic Editor

PLOS Computational Biology

Marieke van Vugt

Section Editor

PLOS Computational Biology

Reviewer's Responses to Questions

**Comments to the Authors:**

Reviewer #1: The authors have thoroughly addressed all of my concerns.

Reviewer #2: The authors have addressed all my concerns in the revised manuscript.

**Have the authors made all data and (if applicable) computational code underlying the findings in their manuscript fully available?**

Reviewer #1: None

Reviewer #2: Yes

PLOS authors have the option to publish the peer review history of their article (what does this mean?). If published, this will include your full peer review and any attached files.

Reviewer #1: **Yes: **Cristiano Capone

Reviewer #2: No

---

## [Editor Report · Acceptance letter]

28 May 2024

PCOMPBIOL-D-23-01190R1 

Cholinergic modulation supports dynamic switching of resting state networks through selective DMN suppression

Dear Dr Krishnan,

I am pleased to inform you that your manuscript has been formally accepted for publication in PLOS Computational Biology. Your manuscript is now with our production department and you will be notified of the publication date in due course.

With kind regards,

Lilla Horvath
